# Differentiation of Livestock Internal Organs Using Visible and Short-Wave Infrared Hyperspectral Imaging Sensors

**DOI:** 10.3390/s22093347

**Published:** 2022-04-27

**Authors:** Cassius E. O. Coombs, Brendan E. Allman, Edward J. Morton, Marina Gimeno, Neil Horadagoda, Garth Tarr, Luciano A. González

**Affiliations:** 1Sydney Institute of Agriculture, School of Life and Environmental Sciences, Faculty of Science, The University of Sydney, Sydney, NSW 2006, Australia; luciano.gonzalez@sydney.edu.au; 2Rapiscan Systems Pty Ltd., 6-8 Herbert Street, Unit 27, Sydney, NSW 2006, Australia; ballman@rapiscansystems.com; 3Rapiscan Systems Pte Ltd., Singapore 348574, Singapore; emorton@rapiscansystems.com; 4University Veterinary Teaching Hospital Camden, Sydney School of Veterinary Science, Faculty of Science, The University of Sydney, Sydney, NSW 2006, Australia; marina.gimeno@sydney.edu.au (M.G.); neil.horadagoda@sydney.edu.au (N.H.); 5School of Mathematics and Statistics, Faculty of Science, The University of Sydney, Sydney, NSW 2006, Australia; garth.tarr@sydney.edu.au

**Keywords:** classification, hyperspectral sensors, offal, organ type, short-wave infrared, visible spectrum

## Abstract

Automatic identification and sorting of livestock organs in the meat processing industry could reduce costs and improve efficiency. Two hyperspectral sensors encompassing the visible (400–900 nm) and short-wave infrared (900–1700 nm) spectra were used to identify the organs by type. A total of 104 parenchymatous organs of cattle and sheep (heart, kidney, liver, and lung) were scanned in a multi-sensory system that encompassed both sensors along a conveyor belt. Spectral data were obtained and averaged following manual markup of three to eight regions of interest of each organ. Two methods were evaluated to classify organs: partial least squares discriminant analysis (PLS-DA) and random forest (RF). In addition, classification models were obtained with the smoothed reflectance and absorbance and the first and second derivatives of the spectra to assess if one was superior to the rest. The in-sample accuracy for the visible, short-wave infrared, and combination of both sensors was higher for PLS-DA compared to RF. The accuracy of the classification models was not significantly different between data pre-processing methods or between visible and short-wave infrared sensors. Hyperspectral sensors, particularly those in the visible spectrum, seem promising to identify organs from slaughtered animals which could be useful for the automation of quality and process control in the food supply chain, such as in abattoirs.

## 1. Introduction

The routine differentiation of livestock organs at abattoirs by meat inspectors occurs based on organ type combined with the gross exclusion diagnosis of potential diseases rendering offal safe for human consumption by trained meat inspectors and a supervising veterinarian [1]. This differentiation is a manual-labour-intensive process and includes the potential for human error. Automation of the post-mortem process in abattoirs has been trialled in previous studies using non-contact and non-invasive imaging methods, including x-ray attenuation, computed tomography, and hyperspectral (HS) imaging for livestock body composition analysis [2,3]. Such systems have allowed for meat and organs to not be destroyed or contaminated during analysis, with rapid on-line technologies providing instant feedback at processor chain speed and no need for sample preparation or external transportation [2,3,4]. However, these non-invasive imaging systems generate extensive data, which require pre-processing methods and algorithm development to be sufficiently accurate and efficient for use in industry [2,3,5].

The characteristics, analysis, and applications of spectral imagery in meat quality evaluation were comprehensively reviewed by Elmasry et al. [2], who concluded that HS imaging systems could be used successfully as quality control tools in meat processing industries. Computer vision analysis of conventional red-green-blue (RGB) digital images can differentiate objects based on size, shape, and colour, although textural and chemical differences cannot be detected [5]. Hyperspectral imaging measures the reflectance of light in multiple narrow bands along the light spectrum and has shown great potential in animal industries [6,7]. This allows for spectra to be extracted from each pixel within the image and thereby improves the classification accuracy compared to RGB imaging when differentiating samples that look alike, such as ground meat, by species [8]. These HS technologies can be split into fractions of visible (VIS; 400–900 nm) and short-wave infrared (SWIR; 900–1700 nm) spectra. In the agriculture sector, HS has been used for the prediction of quality, safety, contamination detection, microbial spoilage, and chemical composition of fruits, cereal grains, animal feed and meat [9,10,11,12]. The spectral data downloaded from HS devices are compared between two objects of interest, which can be differentiated based on differing spectral signatures by peaks or differing intensities at certain wavelengths, with classification occurring through the development of prediction models [11].

In the sphere of organs, differentiation and segmentation have occurred based on different spectral intensities of porcine arteries, veins and organs, including the liver and colon [13] and five tissues (peritoneum, urinary bladder, spleen, small intestine, and colon) during open exploratory surgery on a pig [14]. However, these studies were limited by a sample size of one animal, and no further studies have attempted to differentiate animal organs by type. Despite this, studies examining HS sensors have successfully used seven SWIR wavelengths to differentiate offal from lamb muscle [15], the range of 400–1000 nm VIS-SWIR HS to visualise and differentiate ground pork lung from ground pork meat [16], and similarly three VIS and two SWIR wavelengths to differentiate beef from chicken in mince mixtures [17]. In addition, a benchtop spectrometer containing VIS and SWIR sensors was successful in differentiating beef, lamb, pork, and chicken meats from one another [18].

Different data pre-processing and machine learning methods to analyse spectral data are common [19]. However, the comparison of methods is rare in scientific literature, and it is unclear which methods may be superior. For instance, several studies use absorbance instead of reflectance data [20], others use first- and second-order derivatives to capture changes in the spectra [21], and others have combined these with smoothing of the spectra, such as centred moving average, multiplicative scatter correction (MSC), detrending, standard normal variate (SNV) or Savitzky–Golay filtering [22] to reduce non-chemical background and baseline signals from spectra [11,19,23]. Kamruzzaman et al. [24] used a centred moving average of reflectance spectra and found no improvements with derivatives, MSC and SNV, while Kamruzzaman et al. [17] concluded that raw absorbance spectra were optimal.

The aims of the present study were to: (1) investigate the differences between livestock organs in spectral signatures generated from VIS and SWIR imagery; (2) explore the potential of these to differentiate bovine and ovine parenchymatous organs (heart, kidney, liver, and lung); and (3) evaluate the effect of different data pre-processing techniques and machine learning methods on the accuracy of organ classification. Both reflectance and absorbance data, and their first and second derivatives as pre-processing methods, were used as predictors with partial least squares discriminant analysis (PLS-DA) and random forest (RF) algorithms. It was hypothesized that a multi-sensory platform could provide a spectral profile of individual organs that can be used for the development of discrimination algorithms for the automation of this process into food safety and quality control in the red meat industry.

## 2. Materials and Methods

No animals were slaughtered for the purpose of this study, with offal being obtained from an abattoir and a butcher. Therefore, animal ethics approval was not required.

### 2.1. Sample Collection and Scanning Procedure

A total of 104 parenchymatous bovine and ovine organs were collected over two days from a collaborating abattoir and local butcher and maintained at refrigerator temperatures (1–4 °C) prior to scanning (Table 1). The organs included in this study were heart (*n* = 33), kidney (*n* = 20), liver (*n* = 29) and lung (*n* = 22). The sampling was random, and, therefore, breed and production system information was not known because abattoirs do not normally collect such information.

A prototype multi-sensory platform consisting of dual-view multi-energy X-ray attenuation and a VIS and SWIR HS imaging system (Rapiscan Inspection System AK198, Rapiscan Systems Pte Ltd., Singapore) connected to a Cube computer running Ubuntu (Linux OS) was used for the imaging of the organs (Figure 1). Organs were placed in a sealed tray with a transparent acrylic lid to ensure HS penetration and double containment, which was placed within protective lead curtains for scanning. A conveyor transported the samples from end to end (6.64 s for 1260 mm, 189.8 m/s) with both sides protected by lead shielding while X-rays were on. Illumination was provided by a light-emitting diode (LED) strip lamp for VIS and a quartz infrared (QIR) lamp for SWIR.

The HS imaging system consisted of two sensors covering the spectral range from 400 to 900 nm (VIS) and 900 to 1700 nm (SWIR). The VIS (Basler Ace GigE, Photonic Science, East Sussex, UK) and SWIR (Snake A/C GigE v3 AK081, Photonic Science, East Sussex, UK) sensors were powered by 12 V power supply units and fitted with Specim spectrographs (VNIR V10E and NIR V17E, respectively) and a Grade 1 InGaAs detector with air-cooled housing. Spectral resolutions were 3 and 5 nm for VIS and SWIR, respectively, with both sensors capturing 200 spectral slices per second. Exposure time, image size (width, length, offset) and acquisition rate were controlled by Ubuntu (Linux OS) computer programs (eBUSPlayer SDK, Pleora Technologies, Kanata, Canada) and stream2camstodisk command line (B Allman, pers. comm.) in the Aravis environment of Linux. Spectral increment was approximately 1.5 nm between contiguous bands, with 300 bands for VIS and 512 for SWIR. The VIS sensor had a 1920 × 1200 (spectral × spatial) pixel sensor, spectral binned four times and offset 70 pixels, spatial dimension was not binned, and offset was 550 pixels, equalling 300 bands. The SWIR sensor had a 640 × 512 (spatial × spectral) pixel sensor, offset by 64 pixels, and the area captured was 256 pixels. These dimensions were chosen in order to achieve 150 frames per second (fps) for both HS sensors. Exposure times were calculated from 150 Hz, resulting in a 6.666 ms refresh rate, so exposure times were 6.4 ms for VIS and 4 ms for SWIR to download data at this refresh rate.

### 2.2. Extraction and Analysis of Spectral Data

Scanned images (PNG) trimmed to comprise the tray containing organ samples were constructed from 200–400 frames generated by the HS sensors using MATLAB programming language (MATLAB R2021a, Mathworks Inc., Portola Valley, CA, USA). ImageJ (version 1.53a; RRID:SCR_003070) was used to markup regions of interest (ROI) with 7 × 7 pixels in size upon each complete organ image avoiding visible fat and pixels appearing out of focus. Three to eight ROI were marked-up upon each organ depending on the organ’s size, with larger organs having more ROI than smaller organs. Images were viewed in GIMP (GNU Image Manipulation Program version 2.10.18, RRID:SCR_003182), and, subsequently, pixel values of the ROI were obtained, which were then written into a MATLAB algorithm (B. E. Allman, pers. comm.) to obtain reflectance spectra for each image. Spectra extracted from each ROI were visually checked for uniformity with other ROIs within each sample. Output spectral data (VIS and SWIR) were averaged per organ.

### 2.3. Data Processing and Outlier Removal

Mean reflectance HS data per organ were imported into R software [25]. Both VIS and SWIR spectra were subjected to a principal components analysis (PCA) model as per Logan et al. [26]. Each dataset was independently visualised using PCA (Q residuals and T^2^ Hotelling values) with 2 components using the *mdatools* package [27] to detect outliers defined as observations with orthogonal and score distances >20 on the residual plot [28]. Three outliers were detected and removed from VIS and one from SWIR (Table 1). Subsequently, all datasets were trimmed manually to remove machine artifact effects at the start and end of each spectrum, which presented as flat regions. The final spectra for analysis contained wavelengths from 470.5 to 800.5 nm for VIS and 1000.5 to 1600.5 nm for SWIR. A combination dataset (COMB) was created by merging the trimmed VIS and SWIR spectra. To smooth the spectra and avoid spectral noise, trimmed centred moving average equations were used with a window length of 5 and 20% trim for VIS, whereas SWIR and COMB used a window length of 11 and 10% trim. Cubic polynomial Savitzky–Golay filters [22,29] with identical window lengths were also fitted but did not smooth the spectra as effectively as centred moving average and were therefore not considered. Both reflectance (R) and absorbance (A = 1/log(R) as per Lanza [20]) spectra were subsequently pre-processed using first (d1) and second (d2) derivatives, with all these datasets used to develop subsequent classification models of the organs. All spectral datasets (R, Rd1, Rd2, A, Ad1, Ad2 for VIS, SWIR and COMB) were centred and scaled before model development. Data processing was implemented using the *tidyverse* (RRID:SCR_019186) suite of packages [30].

### 2.4. Statistical Modelling

Classification models using spectral data from three datasets (VIS, SWIR, COMB) and six pre-processing treatments (R, Rd1, Rd2, A, Ad1, Ad2) were tuned using leave-one-out cross-validation (LOOCV). The choice of LOOCV was primarily due to the relatively low sample sizes. The PLS-DA and RF methods used the *pls* and *randomForest* functions, respectively, within the *caret* package [31] to differentiate organ type. Model metrics for goodness-of-fit were evaluated using the multi-class summary in the *caret* package [31]. Model tuning was achieved using a number of components (*ncomp*) ranging from 1 to 25 for PLS-DA and the number of variables available for splitting at each tree node (*mtry*) between 300 to 500 for RF [32,33], based on the highest accuracy and the lowest log loss, respectively, on the LOOCV data. Plots for *ncomp* were visually assessed for the minimal *ncomp* to reach the peak in order to prevent overfitting of PLS-DA models. After the optimal tuning parameters were obtained, the final model was run using the *pls* package [34].

Accuracy, precision, sensitivity, specificity and coefficient of agreement (Kappa) were the model metrics obtained by resampling the PLS-DA and RF discrimination models using LOOCV [35,36]. The best model among all datasets with six pre-treatments was selected based on LOOCV accuracy and Kappa for determination of the in-sample accuracy [32]. Following this, the sensitivity, specificity, precision and balanced accuracy were obtained per organ and HS sensor following PLS-DA and RF modelling. Sensitivity corresponds to the inverse of the out of bag error for each organ. Wavelength variable importance (scaled from 0 to 100) of the COMB dataset was determined using the *varImp* function in the *caret* package [31].

Principal components analysis modelling of the three datasets was completed and visualised using the R package *ggfortify* [37,38].

## 3. Results

### 3.1. Reflectance Spectroscopy

Visible reflectance spectra for the four organs are shown in Figure 2a, with the heart and lung showing greater reflectance compared to the liver and kidney across the range between 500 and 850 nm. The liver and kidney had similar spectral signatures throughout the VIS spectrum except for between 500 and 600 nm, where kidneys had slightly greater intensity. Similarly, hearts showed greater intensity compared to lungs between 500 and 600 nm, but both organs showed similar intensity between 600 and 850 nm. Much stronger separation of the spectra occurred in the SWIR region, particularly between 1050 to 1300 nm, where the lung showed greater reflectance than the heart, followed by the liver, and finally the kidney with the lowest intensity (Figure 2b).

### 3.2. Partial Least Squares Discriminant Analysis

Classification results of the different mathematical pre-processing methods using PLS-DA are shown in Table 2. For VIS spectra, absorbance (A) was selected as the best model because it showed 88% accuracy and 84% Kappa on the LOOCV dataset (*ncomp* = 9) for PLS-DA. However, the accuracy of reflectance (R) was similar to A. The in-sample accuracy for A was 96%, with only four samples misclassified (Table 3). The first and second derivatives of R and A yielded slightly lower accuracy compared to the raw data. All hearts were correctly classified using the VIS spectra, along with 18 of 20 kidneys, 27 of 28 livers, and 20 of 21 lungs (Table 3).

For SWIR spectra, the first derivative of absorbance (Ad1) was selected for PLS-DA because of the highest accuracy on the LOOCV dataset (accuracy 92%, Kappa 90%, *ncomp* = 21), although this did not differ from raw A or second derivative of A of the SWIR spectra (Table 2). In-sample accuracy of 98–99% indicated overfitting; therefore, *ncomp* was reduced to six based on a peak when *ncomp* values were plotted by accuracy on the LOOCV dataset, whereupon accuracy was slightly reduced to 88%. Three hearts were misclassified as lungs, four lungs as livers and two livers as kidneys (Table 3).

The combination of VIS and SWIR spectra resulted in very high overfitting, where all datasets resulted in 100% in-sample accuracy and all *ncomp* > 16 (Table 2). Therefore, the *ncomp* was reduced to six to avoid overfitting, giving in-sample accuracies ranging from 79 to 93% (data not shown). The Ad1 treatment was selected as it had the highest LOOCV accuracy (88%), and PLS-DA modelling of COMB data correctly classified all 28 livers, 28 of 31 hearts, 18 of 20 kidneys, and 19 of 21 lungs (Table 3).

Table 4 shows the goodness-of-fit metrics for VIS, SWIR and COMB for all organs. All metrics across organs ranged between 77 and 100%, with VIS producing more accurate results overall than SWIR, being 90% or more across metrics. Hearts and livers were the best-classified organs by VIS and COMB, showing sensitivity, specificity, precision and accuracy above 90%. On the other hand, SWIR showed the greatest accuracy and sensitivity in classifying kidneys.

Variable importance for PLS-DA predictions of each organ type by the combined spectral model (VIS and SWIR using Ad1 pre-processing) is shown in Figure 3. Hearts showed the greatest importance at 540 nm; kidneys at 780 and 1100 nm; livers at 580 and 780 nm; and lungs at 580 to 600 nm. Livers showed greater variable importance peaks than lungs and kidneys, which in turn were generally higher than hearts, although hearts tended to exhibit more stability across the spectra.

### 3.3. Random Forest

For RF, SWIR and COMB had Ad1 selected with the highest accuracy on the LOOCV dataset (all ≥ 85%); however, the accuracy across different datasets showed an accuracy between 68 and 89% (Table 5). Visible HS data had Rd2 selected with the highest accuracy on the LOOCV dataset. The *mtry* ranged from 310 to 490 with no overfitting. In-sample accuracies were slightly less than accuracies on the LOOCV dataset for all spectral data. Misclassification occurred by all spectra for each organ type (Table 6). Table 6 shows overall accuracies to be highest for livers using SWIR and COMB and for hearts using VIS data. Table 7 shows the RF in-sample classification metrics of organ type, where livers were the best classified overall by RF (sensitivity > 89%, accuracy > 90%). However, hearts showed greater sensitivity and equal accuracy and precision than livers by RF using VIS data. Lungs were the least correctly classified organ regardless of spectral data. However, all sensitivities and balanced accuracies were greater than or equal to 80% regardless of the organ.

Combination VIS and SWIR (Ad1 pre-processing) variable importance of RF modelling is shown in Figure 4 on an overall basis rather than by organ types. In comparison to the PLS-DA clearer peaks were seen across the VIS and SWIR spectra at 590 to 600, 740 to 780, 1080 to 1140 and 1180 to 1200 nm.

### 3.4. Principal Components Analysis

The PCA score plots for PC1 against PC2 for Ad1 data were selected for both VIS and SWIR spectra because this series resulted in the highest accuracy. Figure 5 visually demonstrates that the data points cluster together for each organ type. The PCA showed similar results to PLS-DA and RF, where different organs showed different spectral features for identification, and hearts tended to be the most clustered organ with the least overlapping. Using the combination of both HS sensors (Figure 5c) resulted in greater separation by PCA than using VIS and SWIR sensors individually (Figure 5a,b).

## 4. Discussion

The objective of the present study was to explore the potential of VIS and SWIR hyperspectral data to classify parenchymatous organs by type (heart, kidney, liver, and lung), to assess the suitability of different data pre-processing techniques for HS data, and to compare RF modelling with the more conventionally used PLS-DA. This study was an exploration of the use of HS imaging in organ identification and inspection within the meat processing industry, potentially leading to automation and quality control. Results demonstrated that automated classification for organ type could be performed correctly 95% of the time for VIS and 87% for SWIR using PLS-DA without overfitting. These, in addition to the RF accuracy of 85% for automated classification using the combination of VIS and SWIR sensors, highlight the promise for potential uses of a multi-sensory platform in the beef and sheep meat industries. Potential applications include automated animal organ identification and sorting, processing using robotics, and quality assurance replacing tedious manual procedures normally performed manually by meat inspectors and veterinarians [39]. However, it is important to note that the present study dealt with the classification of organs from both sheep and cattle together. The objective of the present study was to differentiate organs independent of origin. However, it is important to note that large-scale processing plants or abattoirs either slaughter one or the other species, whereas smaller ones often slaughter both species. The differentiation of species using HS sensors is a potential avenue of exploration, though it was not undertaken in the present study because it was not the objective.

Both VIS and SWIR sensors produced similar accuracy in classifying organs by type, although VIS was slightly better and more consistent across pre-processing and classification methods. It was expected that SWIR would be superior, given it comprises a good portion of the near-infrared spectrum which is known to be able to detect C-H and N-H bonds [23,40]. Results from Baeten et al. [9] and Kamruzzaman et al. [15] showed SWIR to perform superior to VIS regions in differentiating fruits and agri-food products and the amount of offal present in a meat mixture, respectively. However, these studies used only one HS sensor encompassing both regions and one classification method (PLS-DA). The combination of the two spectral regions in portable and benchtop devices was also promising in studies of meat quality [23,41] and microbial spoilage in fish fillets [10].

Despite the high accuracy for the spectral differentiation of individual organs, these results are to be interpreted with caution due to the small sample size. Larger trials with more samples from several species, breeds and ages are required to build on this pilot study for differentiation of organs within a multi-species abattoir or supply chain, where one or two incorrectly classified organs will not severely affect the model metrics. For instance, Cozzolino and Murray [18] showed very similar results to the present study when differentiating meat by species, although in this case, SWIR and COMB (94–96%) were more accurate than VIS (85%).

The use of such non-invasive devices in the meat processing industry has been hailed for a long period [3]. However, a lack of consistent accuracy, particularly for quality traits, in conjunction with the high costs of installation, has held such technologies back from industry adoption [23,42]. The present study presents a different application for such technologies, whereupon organ differentiation can take place objectively within the abattoir, and the cost of qualified staff could offset the installation and maintenance cost of the multi-sensory platform. Furthermore, such a platform could also add further value to the data collected by predicting chemical composition, quality control, and detection of health issues, as demonstrated in other studies [12,24,43,44]. Although RGB image analysis by computer vision may have been able to differentiate the organs by type in the present study due to their different appearances [8], the potential identification of other non-visible factors such as species, disease, the chemical composition of organs, or even the addition of other organs more similar in appearance may require broader spectral information for successful differentiation.

A novel aspect of the present study was the use of RF as an alternative classification method and its comparison to the conventional PLS-DA and PCA to discriminate organs based on the spectral signature. Random forest is a classification algorithm that has found multiple applications because of its efficiency in handling large datasets and achieving high accuracy [33]. However, decision tree RF modelling has been sparsely used in hyperspectral classification studies of food [7,11,45] in comparison to PLS-DA and linear discriminant analysis (LDA) [2,9,11,42]. Positive results for RF classification were found in the present study, with accuracy compared to PLS-DA being very similar on the LOOCV dataset and slightly lower on the in-sample dataset. For COMB, RF produced greater LOOCV and in-sample accuracies than PLS-DA. Kong et al. [45] found that RF modelling was superior to PLS-DA when classifying rice seed cultivars in the SWIR spectrum. In most previous qualitative studies with HS, PLS-DA has been preferred to RF and other classification methods [11] because PLS-DA provides a combination of partial least squares regression and LDA [43]. When comparing the PCA plots in Figure 5 with PLS-DA and RF results, most organ samples could be visually classified as a specific type through clustering, although the accuracy was lower (60–75%) compared to PLS-DA and RF on the LOOCV dataset (79–96%). It is worth noting that the PCA in the present study was exploratory and not meant for classification as it is unsupervised [7], whereas the supervised PLS-DA and RF methods were used to build robust classification models as per previous studies comparing PLS-DA and PCA [26,46].

The present study provided good overall accuracy (more than 80%) when using VIS and SWIR HS sensors individually and in combination to differentiate organs by type. However, one limitation was the time required to download, markup, extract and analyse the spectral data, all processes that can be automated based on the results of the present study. Despite a rapid scanning time (5–6 s), image sizes of 48 KB for VIS frames and 13 MB for SWIR frames were large, with some scans having up to 1000 frames downloaded per scan, which can be time and space consuming. This excess consumption may require a high-performance computer, and the time required for image download processing may slow the uptake of these automated HS processes in commercial conditions, as any technology would need to be run at chain speed [47]. Similar issues with image file size were reported by Elmasry et al. [2]. However, these limitations could be easily overcome, and the whole process could be fully automated. The small sample size of the present pilot study resulted in no testing of the calibration model against an independent dataset, although LOOCV is an accepted and widely used method for model evaluation in VIS and SWIR HS studies [7,12,48]. The automatic segmentation of an image into “organ” and “background” would allow for organ identification based on shape analysis and for discriminant analysis of each ROI for classification. Such programming would be needed prior to deployment in the industry, and a larger training library is also essential in improving the accuracy of the model.

Other characteristics that could be examined and quantified by HS imaging to assist with the uptake of these technologies include protein, fat, and mineral concentration of organs. Prior studies have investigated the prediction of these parameters with SWIR HS imaging of lamb meat [24], and near-infrared reflectance spectroscopy on different meat products to varying levels of accuracy [20,23]. However, the use of ground meat as opposed to intact meat for the most successful of these studies has similarly slowed the progress of uptake in processing plants [42].

The differences in reflectance and absorbance intensity between organ types arise from differences in the chemical composition, colour, and tissue morphology, which provide a spectral signature to each organ [7,13]. Biel et al. [49] found that of livers, hearts and kidneys, livers had the most protein, P and K; hearts had the most fat; kidneys had the most Ca and Na. In a study on lamb offal nutritional composition, the lungs had significantly more moisture and Fe than the other organs, whereas the heart had more fat, the liver more Zn, and the kidney more Na [50]. This may correspond to the findings of the present study, where hearts and lungs had stronger reflectance than livers and kidneys. However, a study on pork found that Raman spectral reflectance was higher in the heart, followed by the kidney, and lowest in the liver [51], which agreed with the present study.

## 5. Conclusions

The present pilot study showed that visible, short-wave infrared, and the combination of these hyperspectral sensors with wavelengths between 400 and 1700 nm could be implemented in a multi-sensory imaging system to accurately differentiate organ types. In addition, these sensors could have the potential for other future commercial applications at chain speed, such as measuring the chemical composition and detecting organs with defects or diseases. Both sensors had similar accuracy, although the VIS sensor tended to show greater classification accuracy than SWIR or COMB, and the system was very effective at differentiating livestock organs by type with good accuracy and sensitivity. For the purposes of organ differentiation, the present study showed that VIS HS sensors alone could reliably differentiate between organs without the requirement of an additional SWIR sensor, though further studies are required to ascertain this. The PLS-DA algorithms were slightly more accurate for differentiation compared to the random forest, whereas most data pre-processing methods did not provide significant advantages in spectral smoothing or model accuracy, except for centred moving average and first derivative, which were superior to raw spectra, second derivative, Savitzky–Golay smoothing and SNV. Improvements in sample size and in streamlining the analytical process to provide information in real-time could allow such systems to be deployed into the meat processing industry as control tools for authentication of livestock organs by organ type. The value of the system could potentially increase by including other characteristics such as the identification of species and disease, contamination, and other quality control outcomes.

## Figures and Tables

**Figure 1 sensors-22-03347-f001:**
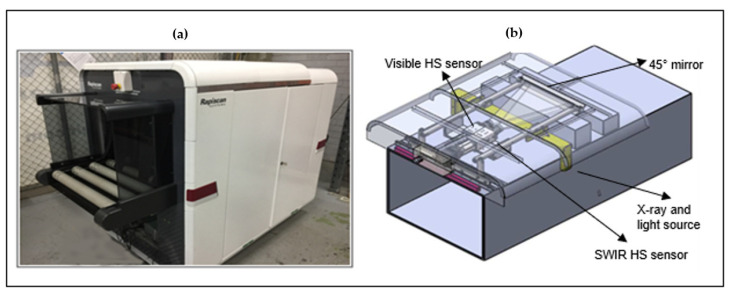
Rapiscan multi-sensory imaging system used to scan livestock parenchymatous organs. (**a**) The external view of the complete prototype imaging system (AK198); and (**b**) A schematic showing placement of the hyperspectral cameras within the imaging system. Source: Rapiscan Systems Pte Ltd., Singapore.

**Figure 2 sensors-22-03347-f002:**
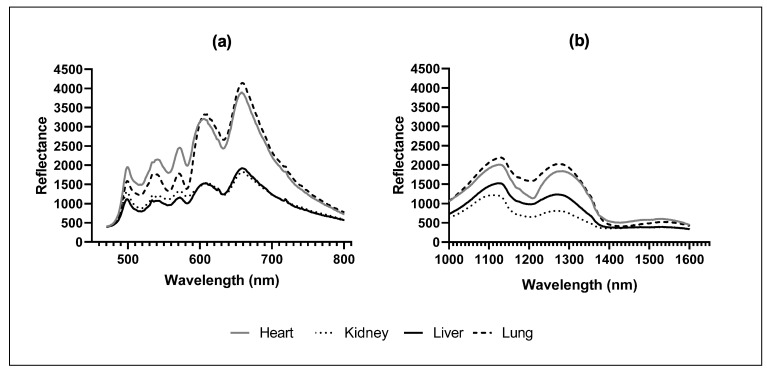
Trimmed centred moving average (**a**) visible (470.5–800.5 nm); and (**b**) short-wave infrared (1000.5–1600.5 nm) spectra for livestock organs by organ type (heart, kidney, liver, and lung).

**Figure 3 sensors-22-03347-f003:**
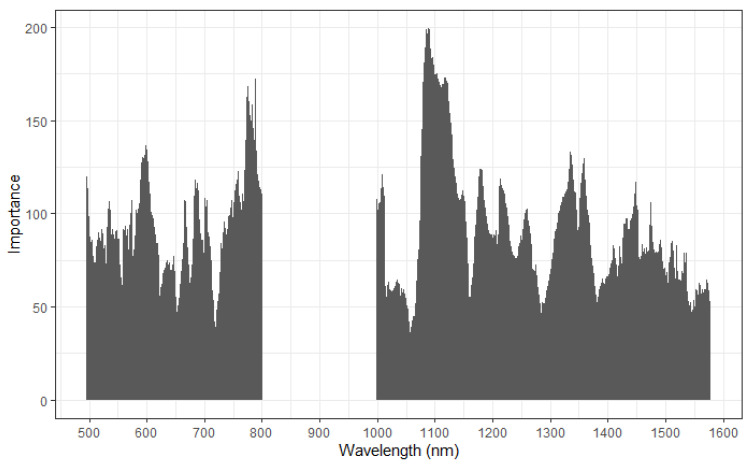
Variable importance for combination of visible (470.5–800.5 nm) and short-wave infrared (1000.5–1600.5 nm) spectra using partial least squares discriminant analysis to identify bovine and ovine organs by type.

**Figure 4 sensors-22-03347-f004:**
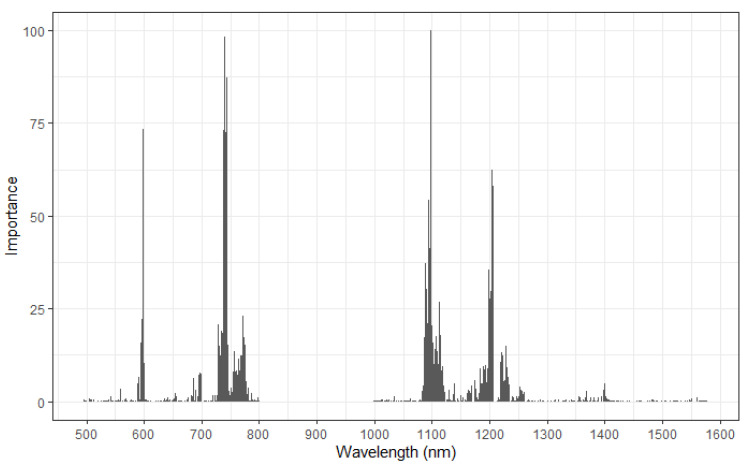
Variable importance for combination of visible (470.5–800.5 nm) and short-wave infrared (1000.5–1600.5 nm) spectra using random forest modelling to identify bovine and ovine organs by type.

**Figure 5 sensors-22-03347-f005:**
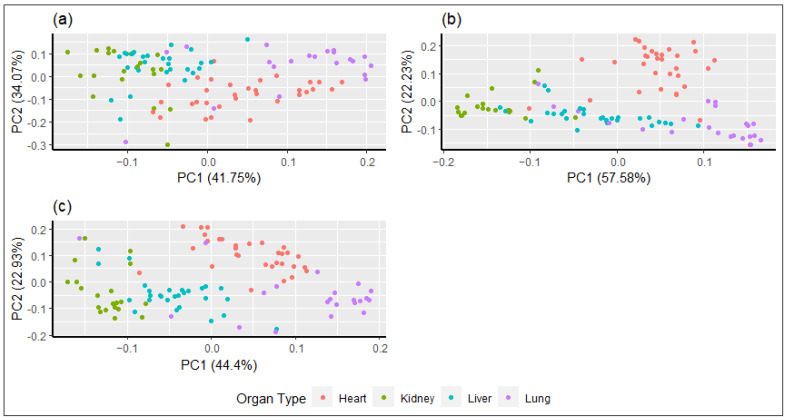
Principal components analysis (PC1 vs. PC2) plots for the first derivative of absorbance hyperspectral data of (**a**) visible (VIS); (**b**) short-wave infrared (SWIR); and (**c**) combination VIS and SWIR spectra for classification of bovine and ovine parenchymatous organs by type.

**Table 1 sensors-22-03347-t001:** Description and number of bovine and ovine parenchymatous organs used to develop automatic identification algorithms from visible (VIS) and short-wave infrared (SWIR) hyperspectral sensors and their combination (COMB) following removal of outliers. One heart from all spectra, one lung and one liver from VIS and COMB spectra, respectively, were removed.

Organ Type	VIS	SWIR	COMB
Heart	32	32	31
Kidney	20	20	20
Liver	28	29	28
Lung	21	22	21
Total	101	103	100

**Table 2 sensors-22-03347-t002:** Partial least squares discriminant analysis (PLS-DA) classification accuracy and coefficient of agreement (Kappa, κ) from visible (VIS), short-wave infrared (SWIR) and combination VIS and SWIR (COMB) hyperspectral sensors to differentiate bovine and ovine hearts, kidneys, livers, and lungs using various pre-processing methods on the leave-one-out cross validation (LOOCV) and in-sample datasets.

Spectra		LOOCV Dataset			In-Sample Dataset	
	*ncomp*	Precision	Accuracy	κ	Accuracy	κ
*VIS*						
R	11	0.87	0.87	0.83	0.96	0.95
Rd1	16	0.85	0.85	0.80	1.00	1.00
Rd2	6	0.77	0.78	0.70	0.90	0.87
**A**	**9**	**0.88**	**0.88**	**0.84**	**0.96**	**0.95**
Ad1	11	0.86	0.86	0.81	0.99	0.99
Ad2	9	0.81	0.82	0.76	0.96	0.95
**Afinal**	**9**	**0.88**	**0.88**	**0.84**	**0.96**	**0.95**
*SWIR*						
R	24	0.91	0.91	0.88	0.99	0.99
Rd1	20	0.91	0.90	0.87	0.99	0.99
Rd2	18	0.91	0.91	0.88	0.99	0.99
A	24	0.92	0.92	0.90	0.98	0.97
**Ad1**	**21**	**0.92**	**0.92**	**0.90**	**0.98**	**0.97**
Ad2	18	0.92	0.92	0.90	0.98	0.97
**Afinal**	**6**	**0.92**	**0.92**	**0.90**	**0.88**	**0.84**
*COMB*						
R	21	0.94	0.94	0.92	1.00	1.00
Rd1	24	0.97	0.97	0.96	1.00	1.00
Rd2	20	0.98	0.97	0.96	1.00	1.00
A	22	0.94	0.94	0.92	1.00	1.00
**Ad1**	**19**	**0.98**	**0.98**	**0.97**	**1.00**	**1.00**
Ad2	16	0.96	0.96	0.95	1.00	1.00
**Afinal**	**6**	**0.88**	**0.88**	**0.84**	**0.93**	**0.91**

*Ncomp*—number of components selected for PLS-DA; R—reflectance; Rd1—first derivative of reflectance; Rd2—second derivative of reflectance; A—absorbance; Ad1—first derivative of absorbance; Ad2—second derivative of absorbance; **bold** indicates the dataset used for reduced *ncomp* and subsequent determination (Afinal or Rfinal).

**Table 3 sensors-22-03347-t003:** Performance of visible (VIS), short-wave infrared (SWIR) and combination VIS and SWIR (COMB) hyperspectral sensors in identifying the type of organs using partial least squares discriminant analysis on the in-sample dataset.

Spectra	Predicted Number of Each Organ			
	Heart	Kidney	Liver	Lung
*VIS*				
Heart	32	1	0	0
Kidney	0	18	0	0
Liver	0	1	27	1
Lung	0	0	1	20
Accuracy (%)	100	90	96	95
*SWIR*				
Heart	28	0	0	1
Kidney	1	19	2	0
Liver	0	1	27	4
Lung	3	0	0	17
Accuracy (%)	88	95	93	77
*COMB*				
Heart	28	1	0	0
Kidney	1	18	0	0
Liver	0	1	28	2
Lung	2	0	0	19
Accuracy (%)	90	90	100	90

**Table 4 sensors-22-03347-t004:** Livestock organ classification from hyperspectral sensors using partial least squares discriminant analysis (PLS-DA) for visible (VIS), short-wave infrared (SWIR) and combination VIS and SWIR (COMB) hyperspectral sensors on the in-sample dataset.

Spectra	Heart	Kidney	Liver	Lung
*VIS* (A)				
Sensitivity	1.00	0.90	0.96	0.95
Specificity	0.99	1.00	0.97	0.99
Precision	0.97	1.00	0.93	0.95
Accuracy	0.99	0.95	0.97	0.97
*SWIR* (Ad1)				
Sensitivity	0.88	0.95	0.93	0.77
Specificity	0.99	0.96	0.93	0.96
Precision	0.97	0.86	0.84	0.85
Accuracy	0.93	0.96	0.93	0.87
*COMB* (Ad1)				
Sensitivity	0.90	0.90	1.00	0.90
Specificity	0.99	0.99	0.96	0.97
Precision	0.97	0.95	0.90	0.90
Accuracy	0.94	0.94	0.98	0.94

(A)—raw absorbance data were selected; (Ad1)—first derivative of absorbance data were selected.

**Table 5 sensors-22-03347-t005:** Random forest algorithm classification accuracy and coefficient of agreement (Kappa, κ) from visible (VIS), short-wave infrared (SWIR) and combination VIS and SWIR (COMB) hyperspectral sensors to differentiate bovine and ovine hearts, kidneys, livers and lungs on the leave-one-out cross validation (LOOCV) and in-sample datasets.

Spectra		LOOCV Dataset			In-Sample Dataset	
	*mtry*	Precision	Accuracy	κ	Accuracy	κ
*VIS*						
R	410	0.72	0.72	0.62	0.69	0.58
Rd1	310	0.80	0.81	0.75	0.80	0.73
**Rd2**	**480**	**0.87**	**0.86**	**0.81**	**0.84**	**0.78**
A	480	0.71	0.72	0.62	0.68	0.57
Ad1	440	0.86	0.85	0.80	0.82	0.76
Ad2	450	0.82	0.81	0.75	0.80	0.73
*SWIR*						
R	320	0.82	0.82	0.75	0.81	0.74
Rd1	500	0.84	0.83	0.78	0.82	0.75
Rd2	340	0.83	0.83	0.78	0.84	0.79
A	330	0.82	0.82	0.75	0.82	0.75
**Ad1**	**490**	**0.86**	**0.85**	**0.80**	**0.84**	**0.79**
Ad2	370	0.83	0.83	0.78	0.83	0.76
*COMB*						
R	310	0.83	0.83	0.77	0.83	0.77
Rd1	460	0.87	0.87	0.82	0.85	0.80
Rd2	440	0.86	0.86	0.81	0.82	0.76
A	390	0.84	0.84	0.78	0.82	0.76
**Ad1**	**450**	**0.90**	**0.89**	**0.85**	**0.87**	**0.82**
Ad2	310	0.86	0.85	0.80	0.85	0.80

*mtry*—number of nodes available for random sampling at each split when developing tree models; R—reflectance; Rd1—first derivative of reflectance; Rd2—second derivative of reflectance; A—absorbance; Ad1—first derivative of absorbance; Ad2—second derivative of absorbance; **bold** indicates the dataset used for final determination.

**Table 6 sensors-22-03347-t006:** Livestock organ classification from hyperspectral sensors using random forest (RF) classification for visible (VIS), short-wave infrared (SWIR) and combination VIS and SWIR (COMB) hyperspectral sensors on the in-sample dataset.

Spectra	Predicted Number of Each Organ			
	Heart	Kidney	Liver	Lung
*VIS*				
Heart	29	1	1	4
Kidney	0	16	2	1
Liver	1	3	25	1
Lung	2	0	0	15
Accuracy (%)	91	80	89	71
*SWIR*				
Heart	27	2	0	0
Kidney	2	17	0	0
Liver	0	1	27	6
Lung	3	0	2	16
Accuracy (%)	84	85	93	73
*COMB*				
Heart	27	1	0	3
Kidney	0	17	0	0
Liver	0	2	27	2
Lung	4	0	1	16
Accuracy (%)	87	85	96	76

**Table 7 sensors-22-03347-t007:** Random forest model metrics per organ for visible (VIS), short-wave infrared (SWIR) and combination VIS and SWIR (COMB) hyperspectral sensors on the in-sample dataset.

Spectra	Heart	Kidney	Liver	Lung
*VIS* (Rd2)				
Sensitivity	0.91	0.80	0.89	0.71
Specificity	0.91	0.96	0.93	0.98
Precision	0.83	0.84	0.83	0.88
Accuracy	0.91	0.88	0.91	0.84
*SWIR* (Ad1)				
Sensitivity	0.84	0.85	0.93	0.73
Specificity	0.97	0.98	0.91	0.94
Precision	0.93	0.89	0.79	0.76
Accuracy	0.91	0.91	0.92	0.83
*COMB* (Ad1)				
Sensitivity	0.87	0.85	0.96	0.76
Specificity	0.94	1.00	0.94	0.94
Precision	0.87	1.00	0.87	0.76
Accuracy	0.91	0.93	0.95	0.85

(Rd2)—second derivative of reflectance data was selected; (Ad1)—first derivative of absorbance data was selected.

## Data Availability

Not applicable.

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
