# Peer review of "Differentiation of Livestock Internal Organs Using Visible and Short-Wave Infrared Hyperspectral Imaging Sensors"

_sensors, 2022, doi:10.3390/s22093347_

Round 1

Reviewer 1 Report

This topic in not new. Lots of studies have been conducted in this regard, and it is challenging to find innovation. Except for Kamruzzaman et al.’s work you mentioned, several other similar recent studies I have found (for examples, another research team, https://doi.org/10.1016/j.saa.2020.119307, https://doi.org/10.1016/j.infrared.2020.103467, https://doi.org/10.3390/foods10092127, https://doi.org/10.3390/foods9020154 ). The manuscript could be improved by discussing the main difference between your work and them.

I have another major concern. Authors used Hyperspectral imaging in classifying different intact Livestock Internal organs. However, conventional imaging could also successfully used in classification. What is the basis for using hyperspectral imaging technique?

Some other specific comments listed below should be addressed.

In M&M section, some details are missing. For example, how many days did you collect the samples? What is the breed of the bovine and ovine organs you used? And so on.

How about the illumination?

What is COMB mean in Table 1?

What is the reflectance unit in Figure 2?

Figures 3 and 4 were not well presented.

Author Response

COMMENT: This topic in not new. Lots of studies have been conducted in this regard, and it is challenging to find innovation. Except for Kamruzzaman et al.’s work you mentioned, several other similar recent studies I have found (for examples, another research team, https://doi.org/10.1016/j.saa.2020.119307, https://doi.org/10.1016/j.infrared.2020.103467, https://doi.org/10.3390/foods10092127, https://doi.org/10.3390/foods9020154 ). The manuscript could be improved by discussing the main difference between your work and them.

RESPONSE: Thank you for your comment. These studies you mentioned focus on adulteration of food products by cooking, leaf lard, offal, and jowl meat, whereas the present study is focusing on the potential installation of HS sensors within a multi-sensory platform in abattoirs. In this case the HS sensors would be used to differentiate different organs by type, which has not been done in prior studies. We have included one of these papers although as per reviewer 2’s comment, we have made our best efforts to keep the discussion and the introduction relatively short.

I have another major concern. Authors used Hyperspectral imaging in classifying different intact Livestock Internal organs. However, conventional imaging could also successfully used in classification. What is the basis for using hyperspectral imaging technique? While we realise that using conventional imaging combined with computer vision may be cheaper, we are testing the accuracy of HS imaging to differentiate organs by type. This is only a pilot study and subsequent studies will cover the use of HS imaging to differentiate based on disease status which is unlikely to be achieved with conventional imaging.

Some other specific comments listed below should be addressed.

COMMENT: In M&M section, some details are missing. For example, how many days did you collect the samples? What is the breed of the bovine and ovine organs you used? And so on.

RESPONSE: All organs were collected over two sampling days and sampled randomly with no information provided about the breeds.

COMMENT: How about the illumination?

RESPONSE: Amended to clarify – X-rays provided general illumination during HS scanning, while VIS illumination was covered by LED and SWIR illumination by QIR.

COMMENT: What is COMB mean in Table 1?

RESPONSE: Amended – combination of VIS and SWIR HS sensors

COMMENT: What is the reflectance unit in Figure 2?

RESPONSE: Reflectance is unitless (amount of light leaving a target to the amount of light striking the target - https://www.l3harrisgeospatial.com/Support/Self-Help-Tools/Help-Articles/Help-Articles-Detail/ArtMID/10220/ArticleID/19247/3377#:~:text=Reflectance%20is%20the%20ratio%20of,is%20called%20%22hemispherical%20reflectance.%22)

COMMENT: Figures 3 and 4 were not well presented.

RESPONSE: Amended figures 3, 4, and 5 to include a border and more break points across the x-axis for clarity.

Reviewer 2 Report

The article is really well written and the work is well conducted, making it clear that the claims made are only based on a limited small dataset and tested with cross-validation only. I would have the following comments

  • You need to provide more information about how the samples are scanned. Organs are not flat, thus, there will be significant variability in focus and out of focus pixels. When you select the “regions of interest”, how do you know that they are in focus?
  • The discussion section is very long, with a lot of material that should either be in the introduction or deleted. Please simplify the discussion section.
  • Line 156 – change to Hotelling’s T2

Author Response

The article is really well written and the work is well conducted, making it clear that the claims made are only based on a limited small dataset and tested with cross-validation only. I would have the following comments

COMMENT: You need to provide more information about how the samples are scanned. Organs are not flat, thus, there will be significant variability in focus and out of focus pixels. When you select the “regions of interest”, how do you know that they are in focus?

RESPONSE: The optics of the multi-sensory platform take care of chromatic effects on focus (set for ~30 mm) and depth of field (~20-60 mm above the conveyor belt). When marking up ROI upon HS images, care was taken to avoid out-of-focus pixels and the extracted spectra were plotted and visually examined to assess uniformity with other ROI from the same sample. Blurring across a uniform region still results in uniform spectral content.

COMMENT: The discussion section is very long, with a lot of material that should either be in the introduction or deleted. Please simplify the discussion section. RESPONSE: Some removed/amended

COMMENT: Line 156 – change to Hotelling’s T2

RESPONSE: Amended

Reviewer 3 Report

In this study were evaluated two methods in classifing animal organs (hyperspectral cameras encompassing the visible and short-wave infrared spectra) and classification models were obtained.

However, some shortcommings remain unclear:

  • L.17 & 99 advert 104 samples, while sum of numbers given in L101-102 as well as in table 1  is less that 104.
  • Fig - 1: remove the paragraph marks and add "a)" which is mentioned in the tilte of the Fig, but not given in it
  • "COMB" is used in table 1, but without explanation of its meanning 
  • Table 3 presents results which are in fawer to VIS and not SWIR & COMB?
  • Table 4, also is the classiofication efficientcy in fawer of VIS, so for all tables this is an important issue which needs to be discussed.
  • Table 6- Accuracy of SWIR is also %
  • Fig 5. what is "d" - under the legend
  • As it is pointed out in the Conclusions - this is a pilot study with its cons and pros. Based on given results, I do not see any need to add SWIR and/or COMB. If thera are any, please point them out, clearly.

Sincerely, 

Author Response

In this study were evaluated two methods in classifing animal organs (hyperspectral cameras encompassing the visible and short-wave infrared spectra) and classification models were obtained.

However, some shortcommings remain unclear:

COMMENT: L.17 & 99 advert 104 samples, while sum of numbers given in L101-102 as well as in table 1  is less that 104.

RESPONSE: Amended to explain which samples were removed as outliers, and the mistake has been corrected

COMMENT: Fig - 1: remove the paragraph marks and add "a)" which is mentioned in the tilte of the Fig, but not given in it

RESPONSE: Amended to add a) and b) to the top of Fig. 1 to avoid going outside the page margins

COMMENT: "COMB" is used in table 1, but without explanation of its meanning

RESPONSE: Amended to spell out abbreviation in table title

COMMENT: Table 3 presents results which are in fawer to VIS and not SWIR & COMB?

RESPONSE: This is correct, although it is worth mentioning that when selecting the models based on accuracy upon the LOOCV dataset, SWIR (92%) had greater accuracy than VIS (88%) (Table 2). Regarding the in-sample dataset (for validation) in Tables 3 and 6 (PLS-DA and RF, respectively), SWIR outperformed VIS for correct classification of kidneys, meanwhile COMB outperformed VIS for correct classification of livers. This has also been mentioned in the results section.

COMMENT: Table 4, also is the classiofication efficientcy in fawer of VIS, so for all tables this is an important issue which needs to be discussed.

RESPONSE: L239-243 (after Table 3) this is mentioned.

COMMENT: Table 6- Accuracy of SWIR is also %

RESPONSE: Amended

COMMENT: Fig 5. what is "d" - under the legend

RESPONSE: There is no “d” for Fig. 5 – this may have been removed by the editor upon returning the manuscript. Figure 5 has been re-done for clarity as well, as per Reviewer 1's suggestion.

COMMENT: As it is pointed out in the Conclusions - this is a pilot study with its cons and pros. Based on given results, I do not see any need to add SWIR and/or COMB. If thera are any, please point them out, clearly.

RESPONSE: Given the small size of the present study, we suggest retaining SWIR and COMB for future studies. This is particularly because we have encouraged the exploration into differentiation of organs based on defects and species. Table 2 shows that on the LOOCV dataset (calibration), there was almost no difference between VIS and SWIR. Consequently, the conclusion has been amended to show the particularly high accuracy of VIS. Using only one sensor in a multi-sensory platform will be cheaper all around but more studies are to be conducted prior to concluding this.

Round 2

Reviewer 1 Report

The authors did not show a good attitude towards revising the paper. The number of words and time you revised may not be as many as the number of words proposed and time spent by reviewers. First of all, I don't see any improvement in the novelty of the article in your revised manuscript. Authors argued their work is innovative that HS sensors would be used to differentiate different organs by type. You only changed a similar object, how about the modeling methods, preprocessings methods, PCA, and so on.

I recommended in round one that conventional imaging could also successfully used in this topic. What is the basis for using hyperspectral imaging technique?

Authors argued that they were testing the accuracy of HS imaging to differentiate organs by type. You said this was only a pilot study and subsequent studies would cover the use of HS imaging to differentiate based on disease status which is unlikely to be achieved with conventional imaging.

Are there any results related to disease status evaluation in this manuscript? This manuscript only focuses on a simply classification.

Did you clarity the differences between conventional imaging and HSI in your Introduction?

At least I didn't see the practical value of HSI compared to conventional imaging from this current work.

As for the breeds, you said samples were randomly with no information provided about the breeds. Then, how do you know whether the HS information you used in classification comes from organ types or different breeds?

As for the reflectance, didn’t you used the HS images calibration by reference panels? The relative reflectance(%) should be more useful. The amount of light striking (you said) may be significantly influenced by the illumination, power stability, mechanical structure, etc, at that time.

Figures? What is content in Figure 5d? Figure 3? The x-axis ranged from 800 to 160?

There are also many problems, I will not list them all.

Reviewer 3 Report

Corrections have contributed to the clarity and clarity of the presented study and the paper is now in a form acceptable for publication.

Author Response

We thank you for your helpful comments!